# Unlocking the Full Sustainability Potential of School Buildings by Reconciling Building Properties with Educational and Societal Needs

**Oskar Seuntjens [1,\*], Matthias Buyle [1]** 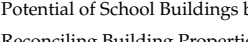**, Bert Belmans [1,2] and Amaryllis Audenaert [1]**

[1]   Faculty of Applied Engineering, University of Antwerp, Groenenborgerlaan 171, 2020 Antwerpen, Belgium
[2]   Faculty of Design Sciences, University of Antwerp, Prinsstraat 13, 2000 Antwerpen, Belgium
**\***   Correspondence: oskar.seuntjens@uantwerpen.be

**Abstract:** This study explores how school buildings can be exploited more efficiently in the future, since, at present, they remain unused for a substantial amount of time. One possibility to tackle this inefficiency, is to involve the local community more closely in usage of its school building. First, a theoretical analysis was carried out to increase the fundamental understanding of the underlying dynamics related to opening school infrastructure to the local community. Second, focus group discussions were organized to research whether involving the local community in the school building was compatible with educational needs. The first highlighted that more extensive building usage could lead to positive social, environmental, educational and economic benefits. In the second, educational experts stressed that they wanted to adopt more innovative and flexible forms of teaching in the future, such as team teaching. Technical directors expressed concerns on safety issues if the local community is to be more closely involved. In the final step, all findings were translated into their technical consequences. From this analysis, it could be concluded that a school building with a high degree of short-term flexibility was the preferred option to reconcile societal and educational needs.

**Keywords:** school buildings; flexible education; adaptable buildings; flexible building usage; school–community partnerships; local community; focus group discussions; causal loop diagram

## 1. Introduction

Dealing with a growing population, increasing social tensions, global warming and resource scarcity are just a few challenges for the next decades. Although these problems require a global approach, school buildings are in a strategic position to address at least some of these issues at the local level. Nowadays, their function is mostly limited to education. Still, they have the potential to form the heart of a community, while, at the same time, reducing environmental impacts of the built environment and increasing revenues for the school management.

In recent years, several countries have developed policy plans with an eye to increasingly opening up school infrastructure to the community, e.g., 'Building Schools for the Future' in the United Kingdom and the 'Masterplan Scholenbouw 2.0' in Belgium [1,2]. The main idea of school–community partnerships is that schools expand their educational mission by involving the wider community in their daily operation. In concrete terms, members of the community, like families or non-profit organizations, would be more intensively involved in using a school building, both during and after school hours. Although school–community partnerships can take different forms, i.e., full-service community schools, full-service schools, family and interagency collaboration and community development [3], they all have the fact that they offer added social value to the local community in common.

In addition, assigning a broader function to schools, by exploiting them more extensively, is also beneficial from environmental and economic points of view. In the context of a growing population and rapid urbanization, there is an increasing demand for public



infrastructure [4]. This need can be met by either building new infrastructure or by using existing infrastructure more extensively. The former solution is costly and has a high environmental impact. The latter implies a more intensive usage, which requires more qualitative and multifunctional infrastructure. In turn, this multifunctional infrastructure can be used more extensively, which is in line with the principles of the circular economy, which advocates using all materials and components to their maximum value [5]. Furthermore, this also reduces the demand for additional infrastructure, resulting in a lower economic and environmental impact for society. Finally, schools can increase their revenue streams by letting school infrastructure to external users.

Although it has been proven that there are several advantages to using school infrastructure more extensively, the instances of external usage after school hours remain limited to date. For example, a study carried out in 2020 by AGION [6] indicated that only 62% of the schools in Flanders (Belgium) were used after school hours, and only for approximately 17 hours, on average, per week. Even when school buildings were used more extensively, this was mostly limited to letting sport infrastructure and multifunctional spaces such as cafeterias. As a consequence, the largest part of the school infrastructure remained unused for a substantial amount of time. In addition, there has been no increase in external usage since the previous study caried out in 2013 by AGION [7]. One of the main reasons for schools to limit use after school hours, is the lack of adequate infrastructure. The study carried out by AGION [6] concluded that schools, where the infrastructure did not allow for opening buildings in a safe way, showed up to 54% less external usage than those schools where the infrastructure could facilitate external usage safely. In this context, 'safe' mainly refers to access to the building. For example, external users should only be granted access to the spaces they intend to use. Secondly, an extensive usage of school infrastructure also implies the possibility of a more flexible building usage. While school buildings would be used as educational facilities during the day, they could be used for other purposes after school hours. This creates an additional layer of complexity, as this can result in fluctuating building requirements, e.g., other dimensions of rooms or other requirements of technical installations.

Therefore, in order to facilitate and maximize flexible building usage, it is crucial that the schools of the future be more adaptable. However, given the core mission of schools, it is fundamental that the after-school use is compatible with the educational vision of the school. From the past, we can observe that innovative architecture can offer solutions to create spaces that can accommodate multiple functions in a sustainable manner. It has been stressed by Barrett et al. [8] that the infrastructure and physical characteristics of learning spaces can have a significant impact on the educational progress of students, and that each type of education has specific building requirements. So, in order to enable more extensive use of school buildings in a sustainable way, it is important to identify the latest developments in educational visions. Examples of such educational innovations are more interactive teaching methods, blended learning and team teaching, which have already proven their pedagogical value [9–11]. Furthermore, it must be kept in mind that educational visions are dynamic. Therefore, the building must be sufficiently adaptable in order to facilitate educational visions that may be unknown today [12]. The building characteristics that are ultimately chosen, with respect to the educational vision, will, in turn, have an effect on whether, and to what extent, a school building can be opened up to external users.

To summarize, there are many advantages to using school infrastructure more extensively. However, to date little is known about how school infrastructure can facilitate this extensive, and, thus, flexible, building usage. In this context, the main goal of this study is to research how to fully exploit the potential of school buildings in the future. The Materials and Methods section describes the used research methodology more in detail. In general, the research methodology consists of three steps. First, the dynamics of opening up school infrastructure to external users from a broader societal perspective, approached from a theoretical point of view. Second, the future needs of school infrastructure regarding opening

school buildings to external users and educational visions in a Belgian context, identified through a series of focus group discussions. Third, these identified needs translated into their technical consequences. The findings of these subtasks are displayed in the Results section. The Discussion section elaborates on the results by highlighting possible scenarios towards a more optimal usage of school buildings in the future. Research limitations and research opportunities are discussed as well.

## 2. Materials and Methods

In order to formulate an answer to the general research question, a research structure, which is illustrated in Figure 1, is followed, that consists of two separate steps. First, a theoretical analysis was carried out in order to increase the fundamental understanding of the underlying dynamics related to opening school infrastructure to the local community. Second, the needs of the educational field were identified by means of focus group discussions. These needs were, in turn, translated into their building requirements. The discussion section elaborates on the meaning of the results.

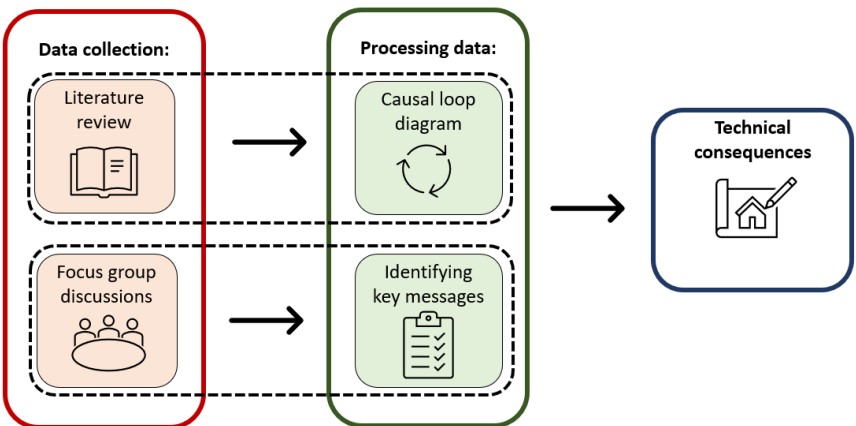

**Figure 1.** Research model.

### 2.1. Theoretical Analysis

To understand the underlying effects of involving the local community in a school building, a theoretical analysis was carried out. It also demonstrated, by means of a causal loop diagram, how these effects were connected to each other. The consequences of involving the local community in the school building were identified through a literature review. Web of Science and Google Scholar were used as bibliographic databases using the following keywords for a first screening: 'school community partnerships', 'future schools' and 'circular buildings'. After analyzing these sources, and the sources to which they referred, a selection was made of sources that discussed the consequences of using school buildings more extensively. In addition, the model was further enriched by effects that could be derived from basic economic theory. Some possible pathways were also proposed by the authors that may enhance certain effects. In the next step, all these identified consequences were subdivided according to the type of effect, i.e., social, educational, environmental and economic. Moreover, all the effects were placed on a two-dimension axis. On the horizontal axis, a distinction was made between short-term and long-term consequences, since many effects related to education only manifest themselves over time [13], while other effects, such as gaining revenues, occur more rapidly. On the vertical axis, effects were categorized according to their impact level. While some consequences occur at school level, there are also consequences that affect the whole society. A clear overview of the link between the different effects can be found in https://doi.org/10.34934/DVN/VPAFBQ (accessed on 20 September 2022).

*2.2. Focus Group Discussions*

It has been stated by Heidrich et al. [14] that it is crucial to understand the needs of stakeholders when designing a building that must facilitate flexible usage. In the case presented here, the study focused on whether extensive school building usage was compatible with current and future educational needs. To study to what extent this was possible, one type of focus group discussion was carried out twice with stakeholders from the educational field. The first focus group consisted of nine participants, who were linked to technical and vocational education, two educational experts, who support teachers and try to implement new teaching methods in the long run, five technical school directors, who manage their school buildings, and two civil servants, administering several schools within the same educational network. As the way of teaching can differ profoundly between technical and vocational schools, on the one hand, and general schools, on the other, a second focus group discussion was organized with three educational experts affiliated with general primary and secondary education. Furthermore, it is important to mention that all participants were involved in the Belgian educational system, as educational visions can vary between regions [15]. The main objective of this qualitative research method was to explore how these stakeholders would (like to) use schools in the future. Furthermore, the intent was also to have different profiles discussed as to whether conflicting interests could, nevertheless, be reconciled. For these reasons, it was decided to use focus group discussions, since they have the advantage of being able to generate a lot of creative answers as discussion between participants is central to this qualitative research method [16].

Both focus group discussions were organized following the same procedure. First an engagement question was asked to make the participants feel comfortable. Next, three exploration questions and one exit question were put forward. An overview of these questions is given in Table 1. The first exploration question was unambiguous and still open-ended to stimulate a discussion between the participants. In the final two exploration questions, the participants were challenged with design exercises. Based upon the discussion from the first exploration question, the participants were asked to draw their ideal school of the future, or at least a part of it. This could be either a detailed or a conceptual design. Despite the participants having no background in building design, this approach offered two advantages. First, it is crucial that the design of a building is based on the needs of stakeholders [17]. By carrying out design exercises, it becomes possible to explore how the educational stakeholders would translate their needs into spatial implications. Second, by adding a spatial component to the focus group discussions, the participants were forced to think out of their comfort zone, which could provide additional insights. The difference between the two design exercises was that for the first one the participants were asked to draw a floorplan, while for the second exercise they had to draw a cross section of their ideal school to trigger the participants to take a different view. Afterwards, the results of the exercises were discussed with the other participants. In the first focus group discussion, the design exercises were carried out in pairs. It was decided to pair people with identical backgrounds. It would then be possible to distinguish whether different backgrounds would also result in different designs. In the second focus group discussion, the three participants carried out the design exercises by themselves as they were all educational experts. An overview of the results from the focus group discussions can be found in https://doi.org/10.34934/DVN/VPAFBQ (accessed on 20 September 2022).

**Table 1.** Question guide focus group discussions.

| Engagement question | Which school are you affiliated with and what is your position? |
|---|---|
| Exploration question 1 | What would your ideal school look like in the future? |
| Exploration question 2 | Can you make a design of your ideal school (floorplan)? |
| Exploration question 3 | Can you make a design of your ideal school (section)? |
| Exit question | Is there anything else you would like to mention about the school of the future? |

The focus group discussions were recorded by using a voice recorder, which was, in turn, used to transcribe and clean the collected data. These data were processed in two steps. First, a qualitative content analysis was carried out in order to identify the key messages of the participants. These key messages could include both opportunities and threats for the school of the future. It was also examined whether participants with different profiles had different priorities. Second, all identified needs and concerns were translated into their technical consequences. In order to address these consequences, several infrastructural measures were proposed.

## 3. Results

The results section is divided into three components. First, the theoretical analysis regarding the involvement of the local community in a school building is discussed and illustrated in Figure 2. In the second part, the main results of the focus group discussions are addressed and summarized in Table 2. Finally, it is examined how these results impact the infrastructure of school buildings. A comprehensive overview of these findings can be found in Table 3.

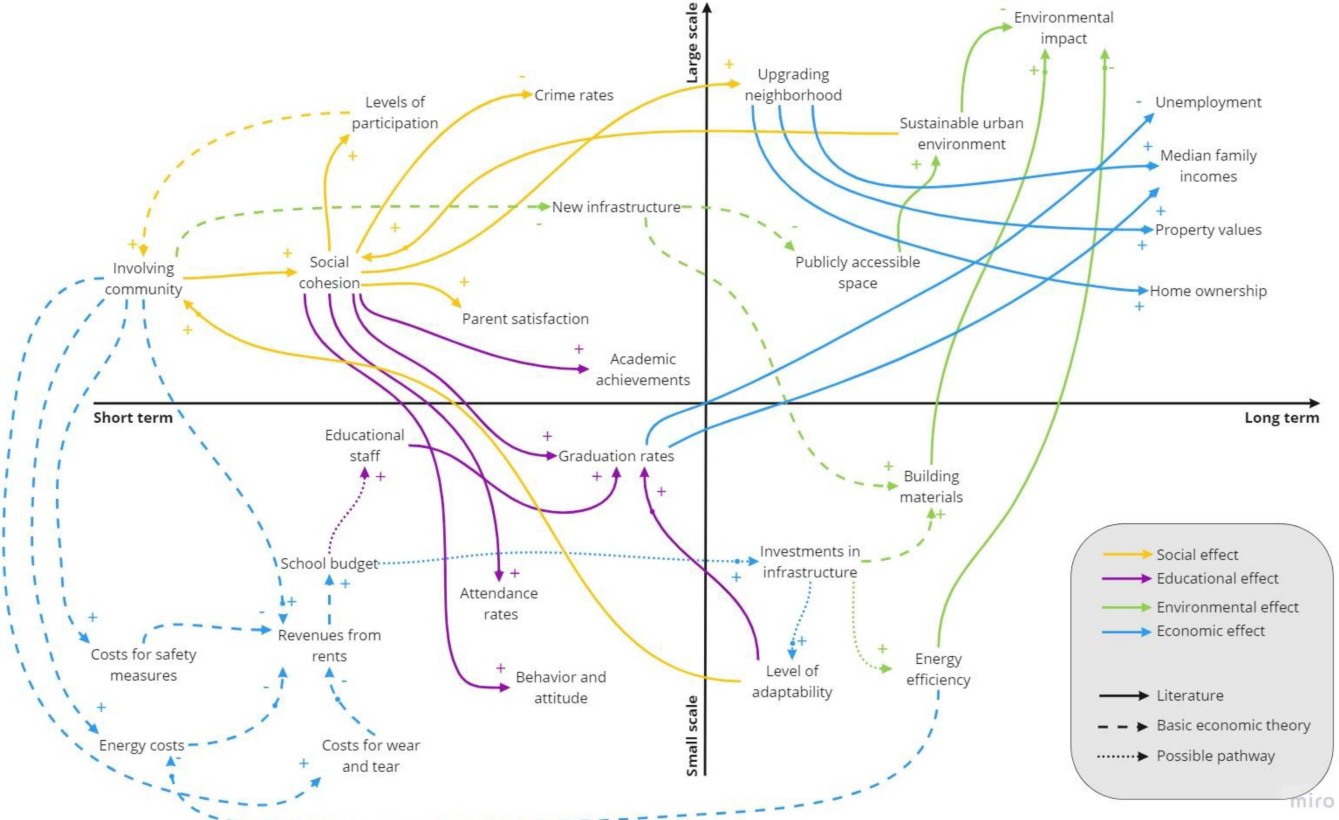

**Figure 2.** Causal loop diagram extensive school building usage.

**Table 2.** Key messages focus group discussions. Legend for symbols: + = Positive impact; ++ = Very positive impact; - = Negative impact; – = Very negative impact; Blank = Not relevant.

| Type of Stakeholders | Opportunities | Impact | | | |
|---|---|---|---|---|---|
| | | *Social* | *Educational* | *Environmental* | *Economic* |
| Educational experts | Provide more flexibility in education | | ++ | | |
| Educational experts | Making the transition towards team teaching | | ++ | | |
| Educational experts & technical directors | Involving the community in education | ++ | + | ++ | |
| Technical directors | Generating revenue by letting infrastructure | | | | ++ |
| **Type of stakeholders** | **Threats** | | | | |
| Educational experts | Adequate infrastructure is required to facilitate more innovative forms of education | | – | | |
| Technical directors | Opening school infrastructure to external users can lead to safety issues | - | | - | – |

**Table 3.** Stakeholder needs vs. technical consequences. Legend for symbols: ✓ = Required; ✓✓ = Strongly required; – = Strongly not recommended; - = Not recommended; + = Recommended; ++ = Strongly recommended; Blank = Not relevant.

| | | Requirements | | |
|---|---|---|---|---|
| | | *Short-Term Flexibility* | *Long-Term Flexibility* | *Safety* |
| **Education type** | Traditional teachingFlexible teachingTeam teaching | ✓✓ ✓ | ✓ ✓✓ | |
| **Extensive usage** | Involving communityStructural external usage | ✓✓ | ✓ ✓✓ | ✓✓ ✓ |
| | | **Infrastructural measures** | | |
| **Floorplan layout** | Fixed layout | – | - | + |
| | Open space plan | ++ | + | |
| | Spaces with fixed functions | – | | |
| | Multifunctional spaces | ++ | | |
| | Moveable walls | ++ | | + |
| | Demountable and reusable walls | | ++ | |
| **Structure** | Oversizing structural components | | ++ | |
| | Bearing wall structure | | – | |
| **Organization** | Circulation inside building | | | - |
| | Circulation outside building | + | | ++ |
| | Zoning | | | ++ |
| | Storage | ++ | | ++ |
| **Technical services** | Conventional HVAC | – | - | - |
| | Adaptable HVAC | ++ | + | + |

## 3.1. Theoretical Analysis Extensive School Building Usage

As shown in the causal loop diagram displayed in Figure 2, opening school infrastructure to the local community has the potential to induce a chain reaction of consequences. These effects are of various kinds and can occur at different time horizons and scales. To

begin with, involving the local community in a school building can lead to multiple positive social consequences, shown by the yellow arrows. To begin with, Stefanski et al. [18] argue that involving the broader community in the school enforces its social cohesion. In Sandtown-Winchester, a neighborhood in Baltimore, Proscio [19], it was found that school–community partnerships led to a decrease in crime rates in the long run. In this case, the revitalization of the neighborhood also resulted in increasing property values and homeownership. However, this stronger social cohesion could also result in shorter term social benefits. For example, Millsap et al. [20] evaluated a case study in Detroit and noted an increase in parental satisfaction while other studies have measured higher levels of trust and participation [21,22]. The latter would create a positive feedback loop because it would ensure that the community was even more intensely involved in the school building. In turn, this would strengthen all the social effects.

Opening school buildings to the local community also has an impact at the educational level, illustrated by the purple arrows. In a study from 2007, Whalen [23] noted that the Chicago's community school initiative resulted in improvements in the behavior and attitudes of the students. This was also confirmed by Leone and Bartolotta [24], while Proscio [19] found that the student attendance rate rose from 80% to 94%. Other studies have measured higher graduation rates and better academic achievements. For example, data from 171 schools that took part in the Communities in Schools initiative showed that 76% of the students improved their academic performance, that 86% of the eligible students graduated and that the dropout rate was only 4% [25]. Similar results can be found in the studies from Kirkner and O'Donnell [26] and Krenichyn et al. [27]. All these positive educational effects propagate themselves in socio-economic benefits in the long term, such as lower levels of unemployment and higher median family incomes [19,28].

The consequences regarding the environment mainly manifest themselves in the longer term, and are shown by the green arrows in the causal loop diagram. First, as can be derived from basic economic theory, using existing infrastructure more extensively would result in a lower demand for new public infrastructure. This avoids the energy-intensive production of new building materials, resulting in a lower environmental impact [29]. In addition, a lower demand for new infrastructure could also ensure that publicly accessible space was not being compromised. Publicly accessible space is necessary for creating a safe, viable and sustainable urban environment [30]. It is also an adequate approach to mitigate the risk of water flooding in urban environments [31]. Furthermore, it has been stated by Miller [32] that sufficient publicly accessible space is also crucial from a social point of view, as it allows diverse social groups to meet and interact.

Following basic economic theory, using a school building more extensively can also induce short-term economic effects at the school level. These effects are displayed by the blue arrows in the causal loop diagram. In the first instance, operational costs, such as energy costs, costs for safety measures and costs for wear and tear would increase. However, these costs could be compensated by letting school infrastructure to external users. This additional source of income could be used by the school in several ways. A possible choice might be to recruit additional educational staff. This might allow schools to reduce the number of students per class which has a positive impact on the students' performance as highlighted in the results from the STAR project [33]. Another possibility is to use these financial resources to invest in school infrastructure, e.g., by increasing the energy efficiency of the school building, resulting in lower energy costs and a reduced environmental impact. Besides energetic renovations, the adaptability of the building could be increased. This could also be interesting from an educational point of view. It has been shown by Barrett et al. [8] that the infrastructure played a significant role in the learning process of students. By making this infrastructure adaptable, it could facilitate different forms of education and, thus, enhance student performance. In addition, increasing the adaptability made it possible to use the infrastructure in multiple ways [34], allowing the community to be involved in the building on a larger scale. This would strengthen all the aforementioned consequences as it creates another positive feedback loop.

To conclude, involving the local community in the school building creates a wide range of consequences. It is important to note that the magnitude of some of these effects depend on which choices a school management makes. Several findings can be derived from the causal loop diagram. However, when the goal is to use school infrastructure more efficiently, two possible pathways can be distilled. The first aims to involve the local community as closely and intensively as possible in the school building. This mainly results in positive social consequences in a relatively short term. It is also beneficial from an environmental point of view, as the demand for new infrastructure decreases and publicly accessible space is preserved. This goal can be strengthened by using the income from letting school infrastructure to external users to invest in the infrastructure of the building, i.e., by making it more adaptable in order to involve the local community on a larger scale. The second pathway is based on the idea of facilitating education in an optimal way. The greatest added values of this are the socio-economic consequences that take place in the long term. To achieve this, revenue must be invested primarily in teaching staff and in infrastructure that stimulates the learning process of students.

### 3.2. Results Focus Group Discussions

On the first exploration question, i.e., what the ideal school in the future might look like, the answers varied according to the profile of the participants. The educational experts, both from technical and vocational education and from general education, agreed that manners of teaching have to change radically in the future. According to them, the traditional way of teaching, where students are given little or no autonomy and do not learn to work independently, is completely outdated and should make way for a more flexible approach to education. An educational expert summarized the problem as follows: "Students in primary school are used to having a lot of freedom, and from the moment they enter secondary school they are stuck in a box". This flexibility can be introduced both at the organizational and the educational level. Regarding the former, the idea was mentioned to give students more freedom to choose at what times they came to school instead of being forced to attend at fixed times. The idea was also raised that students should no longer be divided into classes solely on the basis of their age, but rather on the basis of the skills they have already acquired. With respect to education, the educational experts expressed the belief that it should be possible to use classrooms in a much more flexible way. According to them, it is important to be able to change between different forms of education, since the needs of students may also change. Educational experts affiliated with technical and vocational education elaborated on this, by emphasizing that by providing the necessary flexibility in a building, it would become easier to strengthen the link between theoretical and practical lessons.

In addition to a more flexible form of education, many educational experts also advocated making the transition towards 'team teaching'. Team teaching can be defined as a teaching method where two or more teachers in some level collaborate with each other in the planning, delivery, and/or evaluation of a course [35]. According to the educationalists, this should be accompanied by a transition towards social constructivism, where students are given the opportunity to put their knowledge into practice [36]. This stimulates the students to solve problems by themselves, which, in turn, increases their motivation to learn something [37]. Moreover, it was also stressed that this encourages students to work together, which helps them to acquire soft skills. The biggest advantage, according to the educational experts, is that this learning method allows students to learn at their own pace and they can choose how they want to learn.

When it came to facilitating more flexible teaching and team teaching, all educational experts stressed that the classical idea of a classroom must be abandoned. One expert put it as follows: "Every square meter of a school building, even corridors and lunchrooms, can be used as a place to learn". It was pointed out that, in order to optimally facilitate team teaching, the school infrastructure should be tailored to this teaching method. The results of the design exercises showed how such a building could be conceptually organized. For

example, most of the designs had foreseen a wide variety of spaces in their school building: an instruction room, a space to work in (a) group(s), a low-stimulus room, a media room, smaller coaching rooms, multifunctional spaces and an atrium that interconnected the whole building. This was in stark contrast with the design of traditional school buildings that facilitate a classical teaching method where several smaller classrooms are connected to a corridor.

Besides education, the participants were also questioned about their position on opening school infrastructure to external users. According to the technical directors, this was certainly an option since "a large part of our infrastructure remains unused most of the time". By involving external users more in the school building, these spaces could become more valuable. In addition, the technical directors suggested that this could also allow the schools to generate extra income, which was interesting as they currently lacked financial resources. The technical directors of technical and vocational schools indicated that they would like to do this in a contract-based way by, for example, renting out a number of laboratories or other workplaces to companies that could use these facilities to train their staff. Another example that was given was to open some shops on the school site, e.g., a bakery could be located on the site of a catering school.

Another option was to open school buildings to the broader community in a less structured way. For example, by making spaces available as flexible workplaces or by allowing non-profit organizations to use the school building. The reactions to this proposal varied according to the profile of the participants. The educational experts were in favor, as they felt it could provide added value in the field of education. An example they gave was that the local basketball team could give an initiation in basketball during the sports lessons. An educational expert from technical and vocational education also suggested that experts from industry could also teach students. The technical directors were less enthusiastic, although they recognized the benefits of involving the community in the school building. It was stressed that this could only be an option if it could be organized safely, especially when it came to opening the school building to the local community after school hours. In their own experience, they had noticed vandalism on a few occasions where infrastructure had been opened up to external users. It was emphasized that infrastructure played an important role in this as well. A technical director stated that: "organizing access to the school building is the biggest safety problem". An educational expert elaborated on this by stressing that teachers in primary schools often kept a lot of material in their classrooms and, therefore, did not like to open up their classrooms to external users. These safety-related concerns were also raised during the design exercises. For example, one design provided an external circulation system so that external users could only access the floor where they needed to be and would not have to traverse the entire building. The administrative areas were also strictly separated from the other functions so that they could not be entered by external users under any circumstances. Another design made a strict distinction between a building that could be opened to external users and one that could be closed off completely.

To conclude, all participants envisaged a radically different use of school buildings in the future. The identified key messages, and their impacts, based on the identified categories in the causal loop diagram, are summarized in Table 2 and divided into opportunities and threats. With respect to education, more innovative methods could be used in the future. In particular, switching towards a more flexible form of education and team teaching were favored by the educational experts. However, it was strongly emphasized that this was only possible if the design of the school building was tailored to these teaching methods. In addition, the school of the future should no longer be an isolated island. Involving the local community more in the school building could add value from both an educational and an economic perspective. The technical directors did make the comment that this could only be realized if sufficient safety measures were taken.

*3.3. Design and Technical Consequences*

In this step, the key messages from the focus group discussions were translated into their technical consequences. This allowed us to assess whether the needs of the educational field were compatible with technical and design requirements of local community involvement in school buildings, which, as shown in the causal loop diagram illustrated in Figure 2, induced a range of benefits. First, the identified messages were categorized into two main groups: 'education' and 'extensive building usage'. As far as education was concerned, three different teaching methods were identified. The first was team teaching. All educational experts saw great potential in team teaching. The second was the traditional teaching method. Although many educational experts pointed out that this teaching method was outdated, it was still important to include it, as many schools still adhered to it at present. Therefore, it was interesting to study how this teaching method was compatible with more extensive building usage. The last teaching method was an in between version of the previous teaching methods. The focus group discussions also revealed that there was not one ideal teaching method that was perfect for every student. While a certain group of students could benefit from team teaching, there were also students who benefitted more from a traditional form of education. Therefore, a high degree of flexibility was central to this last form of education, allowing for multiple ways to teach within a single school building. This teaching method was labeled as flexible teaching. Regarding extensive building usage, a distinction was made between a structured and contract-based type of external usage with fixed partners, on the one hand, and a less structured type of external usage, on the other. In the latter case the broader local community was as closely involved in the school building as possible.

All these different types of building usage, both in terms of education and extensive building usage, led to other building requirements. Three perspectives on the different building requirements could be identified: short-term flexibility, long-term flexibility and security regarding opening the school infrastructure to external users. In this context, short-term flexibility referred to the capacity to facilitate different types of building usage within the time span of a day. Long-term flexibility, on the other hand, referred to the capacity to respond to changing needs after a longer period of time. An overview of the relation between the different types of building usage and their building requirements is shown in Table 3.

A number of practical guidelines for designing modern school buildings can be found in the literature, see for example [38–40]. In a state-of-the-art school, which is perfectly designed for team teaching, there is no need for short-term flexibility from an educational point of view. For the predefined flexible teaching method, short-term flexibility is an important building requirement. In this type of education, it is important that teachers be able to teach in different ways and be able to switch quickly between methods. This could be facilitated by making classrooms adaptable in size, for example. Involving the local community in the school building requires a high degree of short-term flexibility as well. This can be explained by the fact that a community can have a wide range of needs. To facilitate these needs, the building must be able to facilitate several types of usage. To enable short-term flexibility, several infrastructural measures can be taken, mainly regarding the design of the floorplan layout. In this respect, an open space plan with multifunctional spaces, that can be reconfigured by using moveable walls, offers many more possibilities than a fixed floor plan with rooms that can only be used to facilitate specific functions. In terms of organization, it is also important to have sufficient storage space in order to apply a clean desk principle. Finally, short-term flexibility also has an impact on the technical services of a school building. Technical services that are demand driven, such as demand controlled ventilation, have a greater potential to facilitate variations in occupancy, both in terms of comfort and energy consumption. Moreover, it is also important that services can react quickly when short-term flexibility is required. Floor heating, for example, is a system that reacts slowly and is, therefore, unable to satisfy a range of needs in a short-term perspective.

In contrast to flexible teaching, which only requires a high degree of short-term flexibility, team teaching and traditional teaching demand a certain level of long-term flexibility. In team teaching, this need translates primarily into the ability to evolve with the latest pedagogical visions. For traditional teaching, this remains limited to being able to respond to demographic waves, i.e., by increasing the capacity through building expansion. This also holds true for involving the local community in the school building. To facilitate structural external usage in schools, long-term flexibility is also important as the clients' needs can evolve over time. Long-term flexibility can be integrated into a building through several infrastructural measures. In contrast to short-term flexibility, the structure of the building plays a major role in facilitating long-term flexibility. By oversizing structural components, such as foundations and beams, future lock-in effects can be avoided. For the same reason, it is also recommended to choose a free plan over a bearing wall structure. Concerning the infill of the building, demountable and reusable internal walls can facilitate a change of the floorplan design in the long term more easily than conventional walls. Finally, it is also important to make the technical services accessible, so that they can be adapted when this is necessary, e.g., when a room changes its function after a couple of years.

The last requirement is related to the safety that the building must guarantee when it is opened to external users. For structural external use this can be more easily guaranteed than when the wider local community is to be included in the school building. This makes sense, as the latter involves many more users who will also want to use the school building in many different ways. To facilitate this external usage safely, some measures can be taken at the organizational level. By choosing external rather than internal circulation, external users do not have to traverse the entire building to get to the place they want to use. From a health perspective, external circulation can be an interesting option as it can prevent the mixing of large groups in a narrow interior space. The latter can lead to infectious problems, as highlighted by the Covid-19 pandemic. If this is not possible, another solution is to work with clear zoning, so that areas that can be opened up for external users are strictly separated from the areas that external users cannot enter under any circumstances. If this zoning is not foreseen in the floorplan layout, moveable walls can help to achieve this. It is also important to provide sufficient storage places for materials so that they cannot be damaged or stolen, which was also emphasized during the focus group discussions. Finally, it is also recommended to use decentral over central technical services. Should there be any problems regarding vandalism resulting in damaged technical services, only one decentral system would have to be repaired, resulting in fewer costs and less disturbance of other users.

## 4. Discussion

Based upon these results, some conclusions can be drawn. First, possible scenarios are discussed that may lead towards a more optimal and efficient usage of school buildings in the future. Later, some guidelines that can be used for further research are proposed.

### 4.1. Future Scenarios

The objective of this research was to explore how school buildings could be used more efficiently in the future. As discussed in the theoretical analysis, two possible pathways were identified. The first aims to provide the best possible education, which results in long term socio-economic benefits. The results from the focus group discussions indicated that switching to team teaching or a more flexible teaching method could be an important step in this direction. However, the educational experts stressed that more innovative teaching methods, like team teaching, require school infrastructure which is completely different from traditional school buildings. In short, school buildings which are perfectly tailored for team teaching require a wide variety of spaces that can be used in different ways and are connected to each other. Furthermore, a high degree of long-term flexibility must be integrated in order to keep up with the latest pedagogical visions. It must be noted that

these types of schools are large in terms of square meters per student and, therefore, have a relatively high cost. In addition, current legislation sometimes makes the size of a school building dependent on the number of students. Therefore, the amount of funding schools receive may be too limited to design a school that is perfectly tailored to team teaching. An alternative is to build or renovate smaller school buildings where a high degree of short-term flexibility in the floorplan layout is integrated. This allows for multiple teaching methods within a single school building.

The other pathway is based on the idea of involving the local community as closely as possible in the school building, resulting in maximal social and environmental benefits. As can be deduced from Table 3, this type of building usage is not entirely compatible with team teaching, as the former requires a high degree of short-term flexibility which is not the case for the latter. Moreover, school buildings which are perfectly tailored to team teaching are also very interconnected. This makes it difficult to control access to the school building. Unlike team teaching, opening school buildings to the local community is compatible with the flexible teaching method.

From this analysis, it can be concluded that a high degree of short-term flexibility can be crucial to reconcile educational and social needs. However, it cannot be ignored that a great number of schools still have a very traditional educational vision. This, in turn, is translated into schools with a rigid structure, which makes it difficult to switch to more innovative teaching methods and to involve the local community. This does not mean that all these schools cannot be used more efficiently in the future. By seeking contract-based partnerships with external users, which require no or only a low level of short-term flexibility, schools can increase their revenue streams. As shown in the causal loop diagram in Figure 2, these incomes could be used to invest in the school building, e.g., by taking infrastructural measures that increase short-term flexibility. This would allow the school, to a certain extent, to switch to a more flexible teaching method and to involve the local community more in the school building, albeit to a limited extent.

### 4.2. Research limitations and Opportunities

The findings of this study can be used as a starting point for the elaboration of future research, mainly in studies that ask for some form of future scenarios on the use of school buildings. This can concern different fields of research, e.g., long-term economic analysis or environmental life cycle assessments. However, it is important to emphasize that this study was carried out on a highly conceptual level. The magnitude in which certain effects manifest themselves is very context-specific. For example, the location (rural versus urban area) of the school has an impact on the effects of involving the local community in a school building. Besides the location, institutional aspects can also limit a school's autonomy in adopting solutions to open its infrastructure to external users. Furthermore, only stakeholders with a background in education took part in the focus group discussions. Including other types of stakeholders, e.g., alterative users or local authorities, might also lead to new insights. In order to exploit the full potential of school buildings in the future, more in-depth research into adaptable buildings is needed. Although a lot of research has been done in this area in recent years, see, for example, Askar et al. [41] and Kamara et al. [42], there are still some blind spots. This is especially the case regarding the potential of technical installations to allow for short-term flexible building usage, which is shown in the work of Seuntjens et al. [43]. If the school of the future is to be used in a completely different, and, above all, much more flexible fashion, it is important that an adequate indoor environment be guaranteed at all times. This is even more important in schools, because, as Mendell and Heath [44] suggest, poor quality indoor environment in schools adversely influences the performance and attendance of students, primarily through health effects from indoor pollutants.

## 5. Conclusions

The results of this study indicate that school buildings are failing to exploit a large part of their potential at present. In particular, a closer involvement of the local community in school buildings could lead to social, educational, environmental and economic benefits in the future. Moreover, these benefits would not only manifest themselves at the school level, but could positively affect the whole of society as well. To achieve this in a sustainable way, this more extensive use must still be compatible with the needs of the educational field. Focus group discussions showed that there is a strong demand for more innovative forms of teaching, such as team teaching, and a more flexible way of teaching. However, these forms of education require a greater degree of adaptability of school buildings, both in the short and long term. Additionally, the stakeholders were in favor of opening up the building to the local community, but the infrastructure must ensure that this can happen in a safe manner. To address these needs and concerns, various infrastructural measures can be taken. In general, it can be concluded that involving the local community is most compatible with the flexible teaching method, as they both require a high degree of short-term flexibility. It is more difficult to reconcile this with a school building that is perfectly designed for team teaching, mainly because it is difficult to open up this type of building in a safe way. Buildings designed for traditional teaching, on the other hand, do not offer sufficient short-term flexibility to meet the wide range of needs of the local community. The results of this study can be used as a starting point for designing future school buildings. In addition, they can also be used as a basis for research that requires future usage scenarios of school buildings.

**Author Contributions:** Conceptualization, O.S. and M.B.; methodology, O.S. and M.B.; validation, O.S.; investigation, O.S.; data curation, O.S.; writing—original draft preparation, O.S.; writing—review and editing, M.B., B.B. and A.A.; visualization, O.S.; supervision, M.B., B.B. and A.A.; project administration, M.B. and A.A.; funding acquisition, M.B. and A.A. All authors have read and agreed to the published version of the manuscript.

**Funding:** This research was funded by the Research Foundation Flanders, grant number 1207520N.

**Institutional Review Board Statement:** Not applicable.

**Informed Consent Statement:** Informed consent was obtained from all subjects involved in the study.

**Data Availability Statement:** Supporting data can be found in the following link: https://doi.org/10.34934/DVN/VPAFBQ.

**Acknowledgments:** We thank Vlaanderen Circulair for supporting the research project "De circulaire school: klaar voor 2050".

**Conflicts of Interest:** The authors declare no conflict of interest.

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
