# Peer review of "Unlocking the Full Sustainability Potential of School Buildings by Reconciling Building Properties with Educational and Societal Needs"

_sustainability, doi:10.3390/su141912136_

Round 1

Reviewer 1 Report

Please, see the document attached

Reviewer 2 Report

The paper addresses an interesting and topical issue within the broader question stated in the title: sustainability potential of school buildings, educational and societal needs. The research question is very precise: whether and how to open schools to the community. Having circumscribed the RQ allows for an original contribution relevant to social, pedagogical and environmental sustainability. The authors discuss the opportunities and risks. The research is well conducted on the literature and with expert and stakeholder consultation methods. Aspects addressed include governance, school management and the characteristics of school buildings in relation to pedagogical trends in different school grades. The working method is rigorous. The reference to school policies in Belgium is interesting. The results of the survey are well presented. 

Precisely in connection with what I have said above, in my opinion some issues could be mentioned in the introduction (paragraph 1), in the discussion of the results (paragraph 4) and in the conclusions that are important to frame the subject.

a) In line 78 I would add a sentence that succinctly reminds us that for some time  architectural design has provided innovative answers to the need to create spaces suited to new organisational and pedagogical models and sustainability. Moreover, many of the suggestions made in section 3.3. Infrastructural consequences, are -I believe- based on architectural literature that is therefore worth quoting. References may be: Nair, P. and Fielding, R. and Lackney, J.A. (2009), The Language of School Design: Design Patterns for 21st Century Schools, DesignShare. Sibylle Kramer (2018), Building to Educate: School Architecture & Design, Braun Publishing AG. Lisa Gelfand, Eric Corey Freed (2010), Sustainable School Architecture: Design for Elementary and Secondary Schools, John Wiley & Sons, 2010.

b) Among the considerations mentioned in lines 484-491, in particular the very context-specific effects, I would also mention the institutional aspects that in different contexts can limit a school's autonomy in adopting different solutions to open its spaces to the community or to other institutions, organisations, businesses in the area. This is a very important fact, more so than a well-designed building.

c) Finally, the issue of safety in the use of the school by the community and other organisations. Mention is made of the problem of vandalism, the need to keep teaching materials safe. A mention should be made (perhaps after line 515) of the health security problems that the covid-19 pandemic has highlighted. In this regard, the solution of direct access from outside for non-school users mentioned in lines 423-424 (external circulation) is interesting and not so obvious even for planners.
